# Salivary IgG4 Levels Contribute to Assessing the Efficacy of *Dermatophagoides pteronyssinus* Subcutaneous Immunotherapy in Children with Asthma or Allergic Rhinitis

**DOI:** 10.3390/jcm12041665

**Published:** 2023-02-19

**Authors:** Jinling Liu, Minfei Hu, Xiaofen Tao, Jing He, Jianhua Wang, Zhenghong Song, Lei Wu, Lanfang Tang, Zhimin Chen, Xuefeng Xu

**Affiliations:** 1Department of Pulmonary Medicine, Children’s Hospital, Zhejiang University School of Medicine, National Clinical Research Center for Child Health, Hangzhou 310052, China; 2Department of Rheumatology Immunology & Allergy, Children’s Hospital, Zhejiang University School of Medicine, National Clinical Research Center for Child Health, Hangzhou 310052, China

**Keywords:** allergen-specific immunotherapy, children, house dust mite, IgE-blocking factor, salivary IgG4

## Abstract

At present, there are no effective, non-invasive, and objective indicators to evaluate the efficacy of pediatric house dust mite (HDM)-specific allergen immunotherapy (AIT). A prospective, observational study was performed in children with *Dermatophagoides pteronyssinus* (*Der p*) asthma and/or allergic rhinitis (AR). Forty-four patients received subcutaneous *Der p*-AIT for 2 years, and eleven patients received only symptomatic treatment. The patients needed to finish their questionnaires at each visit. Serum and salivary *Der p*-specific IgE, IgG4, and IgE-blocking factors (IgE-BFs) were measured at 0, 4, 12, and 24 months during AIT. A correlation between them was also evaluated. Subcutaneous *Der p*-specific AIT improved the clinical symptoms of children with asthma and/or AR. The *Der p*-specific IgE-BF significantly increased at 4, 12, and 24 months after AIT treatment. Serum and salivary *Der p*-specific IgG4 significantly increased with the time of the AIT treatment, and significant correlations between them at different time points were observed (*p* < 0.05). Furthermore, there were significant correlations (R = 0.31–0.62) between the serum *Der p*-specific IgE-BF and *Der p*-specific IgG4 at the baseline, 4, 12, and 24 months after the AIT (*p* < 0.01). The salivary *Der p*-specific IgG4 levels also demonstrated a certain correlation with the *Der p*-specific IgE-BF. *Der p*-specific AIT is an effective treatment for children with asthma and/or AR. Its effect was associated with increased serum and salivary-specific IgG4 levels, as well as an increased IgE-BF. Non-invasive salivary-specific IgG4 may be useful for monitoring the efficacy of AIT in children.

## 1. Introduction

Along with economic development and social progress, the prevalence of allergic disease has increased significantly. In China, house dust mites (HDMs) were the most prevalent allergens in patients with asthma and/or allergic rhinitis (AR) [1]. Furthermore, the severity of rhinitis and asthma was significantly correlated with the skin reactivity to *Dermatophagoides pteronyssinus* (*Der p*) and *Dermatophagoides farinae* (*Der f*) [1]. Currently, allergen immunotherapy (AIT) is the only treatment for patients with IgE-dependent asthma and AR with potential curative effects. When AIT is administered continuously for 3 years, there will still be a permanent clinical benefit for several years after its discontinuation [2].

Sublingual HDM-AIT could induce symptom improvement and immunomodulation [3]. A three-year course of subcutaneous HDM-AIT significantly improved symptoms and quality of life scores in patients with AR [4]. Furthermore, children with AR may obtain better long-term efficacy from HDM-SCIT compared with adults [4]. This effect of AIT on clinical improvement might be associated with reduced allergen-specific IgE levels and elevated allergen-specific IgG4 or IgA levels. There is evidence that AIT can reduce IgE production [5,6]. In contrast, progressively lower serum IgE concentrations were observed after 2 years of immunotherapy and 2 years following the withdrawal of immunotherapy [5]. Additionally, significantly higher IgG4 levels were also observed after the immunotherapy [5]. IgG4 and IgA in both serum and saliva are increased after AIT and may also correlate with the efficacy of AIT [7]. *Der p*-specific IgA antibodies are dominant in the serum and saliva of non-allergic children, while *Der p*-specific IgE and IgG4 antibodies predominate in allergic children, suggesting the specific IgA has a key role in the healthy immune response to mucosal allergens [8]. Salivary *Der p*-IgG4 and *Der p*-IgA levels in AR patients were significantly lower than those in the healthy controls, while *Der p*-IgE levels were significantly higher [9]. Moreover, subcutaneous AIT contributed to sustained increases in *Der p*-IgG4 and *Der p*-IgA in the maintenance phase compared with the up-dosing phase, whereas *Der p*-IgE only increased in the up-dosing phase, further indicating a protective role of IgG4 or IgA against allergies [9].

Due to its competitive ability to bind allergens with IgE, IgG4 is considered an IgE-blocking antibody and a marker of immune tolerance [2]. This IgE-blocking activity can be measured as an IgE-blocking factor (IgE-BF), which can be explained by the allergen-specific IgG4. Our previous study showed that the allergen-specific IgG4 in both saliva and serum significantly increased and correlated during one year of *Der p*-AIT, and their correlation became stronger over the treatment time. Furthermore, the IgE-blocking antibodies induced by the subcutaneous *Der p*-AIT are mainly allergen-specific IgG4, not IgA [10]. Therefore, serum and salivary IgG4 levels can be an alternative immunological marker of AIT. For pediatric populations, a good outpatient experience can greatly increase children’s adherence. Thus, non-invasive salivary IgG4 detection may be used to evaluate the efficacy of AIT. Here, we performed a prospective two-year study to assess the role of subcutaneous *Der p* AIT in children with asthma and/or AR, observing whether salivary IgG4 levels would be an alternative immunological marker of AIT treatment.

## 2. Materials and Methods

### 2.1. Study Subjects

In this prospective observational study, children with asthma and/or AR were enrolled. Mild-to-moderate asthma and AR were diagnosed according to the GINA guidelines and ARIA guidelines, respectively. All the patients were skin prick test positive for *Der p* and had specific serum IgE against *Der p*. All the included patients were followed up to assess AIT within 24 months of recruitment. This study was approved by the Ethic Review Board of Children’s Hospital, Zhejiang University School of Medicine (2016-IRBAL-056). Informed consent was obtained for all the subjects who were under 18 from a parent and/or legal guardian.

### 2.2. AIT Protocol

The standardized HDM-specific subcutaneous immunotherapy was performed at the clinic. The patients received subcutaneous injections of the aluminum-formulated *Der p* Alutard SQ vaccine (ALK-Abello A/S, Horsholm, Denmark). The treatment protocol followed the recommended up-dosing schedule of 16 weeks before reaching a maintenance dose of 100,000 Alutard SQ, with a duration of 24 months for the AIT. The patients were observed in the outpatient for at least half an hour after each administration in order to detect and treat any immediate adverse reaction.

### 2.3. Clinical Evaluations

Patients were requested to assess their symptoms and record the medication use over the past week at each visit during the whole course of treatment. The symptom and mediation use were scored based on WAO recommendations and our previous study [10,11]. Patients needed to assess their symptoms, including nose (blocking, itching, running nose, and sneezing), eyes (itching, hyperemia, swelling, and tearing), and chest (cough, wheeze, chest tightness, and shortness of breath) in the range of 0–3 (0 = no symptom, 1 = mild, 2 = moderate, and 3 = severe). The mean score of nose, eyes, and chest symptoms was calculated individually. The daily score was the sum of all symptoms in a day, and the mean score over 7 days was the score of the week. The overall score about the symptoms in the past week was evaluated using a visual analog scale (VAS), consisting of a 10 cm line ranging from no symptoms (0 cm) to the highest level of symptoms (10 cm). When necessary, the patients were permitted to use rescue medications, which were scored as follows: 1 point was assigned to each dose of antihistamine, antileukotriene, or inhaled β2-agonist; 2 points were assigned for each use of inhaled or nasal corticosteroids; and 3 points were assigned for the use of an oral corticosteroid.

The patients were also asked to fill out the rhinoconjunctivitis quality of life questionnaire (RQLQ) and asthma quality of life questionnaire (AQLQ) in the past week at each visit. The RQLQ is composed of 7 domains, including activity limitation, sleep, non-nose/eye symptoms, practical problems, nose symptoms, eye symptoms, and emotional function, with 28 questions; and the AQLQ included 31 questions involving activity limitations, symptoms, emotional function, and exposure to environmental stimuli, based on previous studies [11,12]. Their scale scores range from 0 (no impairment) to 6 (severe impairment). The sum of the RQLQ and AQLQ is referred to as the total quality of life scale.

### 2.4. Serum and Saliva Samples

Serum and saliva samples were obtained from all the children at 4-time points (months 0, 4, 12, and 24). Serum samples were collected after centrifugation (3000× *g* for 10 min) of blood samples and were then stored at −20 °C until the assays. Unstimulated saliva samples were collected from the oral cavity using standard Salivette devices (Sarstedt AG & Co., Numbrecht, Germany) according to the manufacturer’s instructions. The swab was placed in the mouth for 5 min and then reinserted in the Salivette. Samples were immediately placed on ice and centrifuged at 3000× *g* for 10 min. The supernatants were collected and stored at −20 °C until further analysis.

### 2.5. Serum and Salivary IgE and IgG4 levels to Der p Allergen

Serum and salivary IgE and IgG4 against the *Der p* allergen were measured based on our previous study [10]. *Der p*-specific IgE was detected using the ADVIA Centaur^®^ immunoassay system (Siemens AG, Erlangen, Germany) and was calibrated against ImmunoCAP-specific IgE. The *Der p*-specific IgG4 levels were determined by a sandwich ELISA. In brief, the microplate was coated with 10 μg/mL of a Der p extract (ALK-Abello, Denmark) protein and then blocked with 2% casein and subsequently incubated with diluted standards, controls, and serum samples. Mouse monoclonal antibodies to human IgG4 (ALK-Abello, Denmark) were used at 1/10,000 for detection and revealed by HRP-labeled goat anti-mouse IgG (1/20,000; KPL, Milford, MA, USA) and then followed by Tetramethylbenzidine visualizing. The assays were further calibrated against ImmunoCAP-specific IgG4 (Uppsala, Sweden) by measuring the specific IgG4 concentration in mgA/L of Der p IgG4 in the ELISA standard stock solutions by ImmunoCAP. The lower limit of quantification (LoQ) of serum Der p IgG4 was 3.33 μg/L, and the LoQ of salivary Der p IgG4 was 0.67. Serum samples needed to be diluted at least 5-fold due to the different matrix effects, whereas there was no need to dilute saliva samples.

### 2.6. Allergen-Specific IgE-Blocking Factor

The *Der p*-specific IgE-BF was measured using the ADVIA system (Siemens AG, Erlangen, Germany) based on previous studies [10,12]. In general, the IgE-BF measurement used two different assay procedures. One was for normal IgE determination, including a washing step after serum incubation in order to remove all the serum components except the bound IgE before adding the biotinylated antigen. The other was “competition” IgE determination, excluding the washing step after serum incubation and allowing the non-IgE antibodies to compete with the IgE in binding to the biotinylated antigen. The IgE-BF was defined as “1- (IgEcompetition/IgEnormal)”. In view of the fact that the variation of the IgE measurement was 10%, IgE-BF values less than 10% were considered as not blocking activity.

### 2.7. Statistical Analysis

The continuous variables were expressed as mean and standard deviation or percentile values and compared using the Student’s *t*-test or Mann–Whitney U test. Scatter plots were used to evaluate the correlations between the two variables. All the statistical analyses and graphics were carried out with R software packages (R version 4.0.3). *p* < 0.05 was considered statistically significant.

## 3. Results

### 3.1. Baseline Characteristics of Included Children

Table 1 shows the demographic and baseline data of the AIT patients and control patients. There were no significant differences in age, gender, diagnosis, serum *Der p*-specific IgE and IgG4 levels, and salivary *Der p*-specific IgG4 levels between the two groups. Although asthma symptom scores from the AIT patients were higher than in the control group, other questionnaire data, including the rhinitis symptom score, ocular symptom score, VAS symptom score, medicine score, and rhinitis and asthma quality of life scores, showed no significant differences between them.

### 3.2. Clinical Efficacy Assessment of AIT

The main outcome variables for patients undergoing the two-year AIT include the rhinitis symptom score, ocular symptom score, asthma symptom score, VAS symptom score, medicine score, and rhinitis and asthma quality of life scores. AIT can significantly reduce the VAS rhinitis symptom score, ocular symptom score, asthma symptom score, VAS symptom score, and medicine score at 4, 12, and 24 months compared with the baseline values (*p* < 0.05), which display a continuous downward trend (Figure 1). These results indicated that AIT had a better long-term clinical efficacy for children with asthma and/or AR.

### 3.3. Immunological Responses during AIT

In the AIT group, allergen-specific IgE demonstrated a small increase in *Der p*-specific IgE after 12 and 24 months of treatment (Table 2). However, there were no significant statistical differences compared with the baseline levels. Although *Der f-*specific IgE showed a slight decrease after 12 and 24 months of AIT, there was no significant difference between them. There was no obvious change in any of the tested specific IgEs from the baseline to 12 months in the control group (Table 3). Notably, after the standardizing transform, there was a significant correlation between the serum *Der p*-specific IgE and *Der f*-specific IgE levels at different time points (*p* < 0.001), suggesting a cross-reaction between them (Figure 2).

The *Der p*-specific IgE-BF significantly increased at 4, 12, and 24 months after the AIT treatment (Table 2). However, there were no obvious changes between the baseline and 12 months in the control group (Table 2). Although the AIT treatment only used a *Der p* extract, the *Der f*-specific IgE-BF exhibited an increased trend similar to the *Der p*-specific IgE-BF (Table 2). Additionally, a strong correlation between the *Der f* IgE-BF and *Der p* IgE-BF was also observed at all AIT treatment time points (*p* < 0.05), further indicating their extensive cross-reactivity.

### 3.4. Increased IgG4 Levels during AIT

At the baseline, serum *Der p*-specific IgG4 was detected in all the patients, including the AIT group and control group. After the AIT treatment, all the patients had detectable *Der p*-specific IgG4 at 4, 12, and 24 months. Furthermore, serum *Der p*-specific IgG4 significantly increased with the time of the AIT treatment (Figure 3A). However, no significant changes in the serum IgG4 levels were observed at the baseline and 12 months in the control group. Most notably, the serum *Der p*-specific IgG4 levels after the AIT treatment were significantly higher than in the control group (Table 2 and Table 3).

Different from the serum IgG4 levels, salivary *Der p*-specific IgG4 levels were detected at the baseline only in a small number of patients, regardless of the AIT group or control group. However, the saliva *Der p*-specific IgG4 levels significantly increased after the AIT treatment, especially at 24 months (Table 3 and Figure 3B). Not surprisingly, children with asthma/AR who did not receive AIT had no significant changes in serum or salivary IgG4 levels at the baseline and 12 months (Table 3). Additionally, in the 24th month, the cutoff value of the saliva IgG4 for protective antibodies was 2.006, and its sensitivity and specificity were 0.83 and 0.77, respectively. These results further revealed that AIT treatment could induce individuals to produce *Der p*-specific IgG4, both in serum and in saliva.

### 3.5. Correlation between Serum and Saliva IgG4 during AIT

Figure 3 shows the correlation between serum and salivary IgG4 levels. There was no obvious correlation between them before the AIT treatment (R = 0.19, *p* = 0.98, Figure 3C). However, a significant correlation of *Der p*-specific IgG4 in the serum and saliva was detected at 4 (R = 0.66, *p* < 0.01), 12 (R = 0.41, *p* = 0.02), and 24 months (R = 0.52, *p* < 0.01) of the AIT (Figure 3D–F). This phenomenon was maintained from 4 months to 24 months after the AIT, further suggesting salivary IgG4 levels as real indicators of serum IgG4 levels.

### 3.6. Correlation of Der p-Specific IgE-BF with Der p-IgG4

As the AIT treatment continued, the serum *Der p*-specific IgE-BF also gradually increased with time, and serum *Der p*-specific IgG4 also showed a similar trend, indicating that there could exist a correlation between them. Then, we found that there were significant correlations (R = 0.31–0.62) between the serum *Der p*-specific IgE-BF and *Der p*-specific IgG4 at the baseline (R = 0.54, *p* < 0.01), 4 (R = 0.47, *p* < 0.01), 12 (R = 0.60, *p* < 0.01), and 24 (R = 0.56, *p* < 0.01) months after the AIT (Figure 4). It is noteworthy that the salivary *Der p*-specific IgG4 levels also demonstrated a certain correlation with the *Der p*-specific IgE-BF (Figure 5). Given the fact that salivary IgG4 was detected only in a small number of patients, we did not find significant correlations between the salivary *Der p*-specific IgG4 and serum-specific IgE-BF before treatment (Figure 5A). After AIT began, their correlations gradually increased, and the maximum value appeared at 24 months of AIT (R = 0.38, *p* = 0.025, Figure 5D). These results demonstrated that the effect of AIT on improving clinical symptoms was closely associated with increased IgG4 levels and IgE-BFs.

## 4. Discussion

The present study demonstrated that *Der p*-specific AIT improved the clinical symptoms of children with asthma and/or AR. Its efficacy was closely associated with increased serum and salivary *Der p*-specific IgG4 levels, consistent with an increased IgE-BF. Most importantly, salivary *Der p*-specific IgG4 levels showed the same trend as serum *Der p*-specific IgG4 levels with the treatment time, which can contribute to assessing the efficacy of *Der p-*specific AIT in children.

A large number of studies have confirmed that AIT is effective in patients with allergic diseases [3,4,9,13,14,15]. Moreover, AIT can also prevent the development of asthma in children with rhinitis [15]. Although AIT is a proven therapeutic option for patients with IgE-dependent mild/moderate asthma and AR, there is considerable heterogeneity in the clinical outcomes [2,15,16]. Our study revealed that the patients undergoing the two-year AIT had obvious improvements in the rhinitis symptom score, ocular symptom score, asthma symptom score, VAS symptom score, medicine score, and rhinitis and asthma quality of life scores at 4, 12, and 24 months after the *Der p*-specific AIT treatment. Furthermore, this displays a continuous downward trend with the treatment time, indicating that AIT had a better long-term clinical efficacy for children with asthma and/or AR. In addition to the long-term efficacy for children with AR, single-allergen subcutaneous immunotherapy was also beneficial for AR caused by multiple allergens, including HDMs [17]. These findings further suggested AIT as an effective therapy for mild/moderate asthma and AR. Nevertheless, the potential mechanisms of AIT are not yet fully understood. Currently, non-IgE antibodies, including IgG4 and IgA, are considered to participate in this protective mechanism [2,18].

In general, IgG4 is considered to be a benign, non-pathogenic antibody. Food-specific IgG4 cannot fully illustrate food allergy or intolerance, but it is likely a physiological response of the immune system to food component exposure [18]. IgG4 may help the immune system to inhibit inappropriate inflammatory reactions by competing for allergens with mast cell-bound IgE antibodies or preventing the IgE-facilitated allergen’s presentation, subsequently relieving allergic symptoms [18]. Several lines of evidence have indicated that IgG4 is able to inhibit IgE-dependent events [2]. IgG4 can prevent the binding of high-affinity IgE receptors on the basophils and mast cells and block the cross-linking of allergen/IgE complexes to low-affinity receptors on the B cells, thereby suppressing the IgE-facilitated antigen’s presentation to the T cells, an important driver of allergen-specific TH2 responses [2]. Clinical symptom scores decreased after 4 months and continued to decrease during 12 months of AIT, and the levels of *Der p*-specific IgG4 increased after 4 months and were closely associated with the symptom scores at 12 months of AIT [19]. Our previous one-year study demonstrated that the allergen-specific IgG4 in serum increased over the treatment time [10]. The present study also showed that serum *Der p*-specific IgG4 significantly increased with the time of the AIT treatment, closely associating with improved clinical symptom scores during the two-year treatment. Serum-specific IgG4 levels showed to be a time- and dose-dependent increase in the serum inhibitory activity for both the IgE-BF and IgE-facilitated allergen binding during subcutaneous AIT, suggesting serum IgG4 possibly as a useful alternative marker of the clinical response to AIT [13].

Notably, in the present study, salivary *Der p*-specific IgG4 levels were detected only in a small number of patients compared with the serum IgG4 levels at the baseline. There was no doubt that saliva *Der p*-specific IgG4 levels significantly increased after AIT treatment, in agreement with the trend of improved clinical symptoms. Furthermore, significant positive correlations were observed between serum- and salivary-specific IgG4. The positive correlation between the serum and salivary-specific IgG4 levels to allergens further indicated that IgG4 could be passively transferred rather than locally secreted [8]. Different from this study by Miranda et al. is that they only enrolled those AR and/or asthma children who did not receive AIT [8], while our study focused on the immune responses of children with asthma/AR after receiving AIT. Furthermore, we followed up for 2 years to further study the antibody reactions after AIT. Relative to mechanically stimulated saliva, the salivary IgG4 levels had no significant differences between the resting and mechanically stimulated saliva sample types, suggesting that the saliva sampling method would not affect the test results [20]. Although antibody responses could not be used as a marker of sublingual AIT efficacy at an individual level [21], our findings strongly suggested salivary-specific IgG4 levels as a biomarker for evaluating the efficacy of subcutaneous AIT. This non-invasive salivary IgG4 detection method may greatly increase children’s medical experience, improving patients’ compliance.

The present study had some weaknesses that should be considered. First, because of the difficulty in performing a randomized controlled trial in pediatric populations, we did not carry out a double-blind, randomized, placebo-controlled trial, possibly leading to selection bias. Second, all the patients or their parents were asked to record medications and symptoms at every visit; due to a longer follow-up interval, this could have interfered with the patient’s description of their symptoms. Third, the sample size was relatively small due to the difficulty in collecting pediatric patients and obtaining blood samples. Especially for the control patients who did not receive AIT, it was difficult to obtain blood samples at adequate follow-up intervals.

## 5. Conclusions

We can conclude that *Der p*-specific AIT improves the clinical symptoms of children with asthma and/or AR. The efficacy of AIT was closely associated with increased serum and salivary-specific IgG4 levels, consistent with an increased IgE-BF. Furthermore, salivary *Der p*-specific IgG4 levels showed the same trend as the serum *Der p*-specific IgG4 levels with the treatment time. Therefore, salivary-specific IgG4 measurements may be useful for monitoring its efficacy on children with asthma and/or AR during allergen-specific AIT.

## Figures and Tables

**Figure 1 jcm-12-01665-f001:**
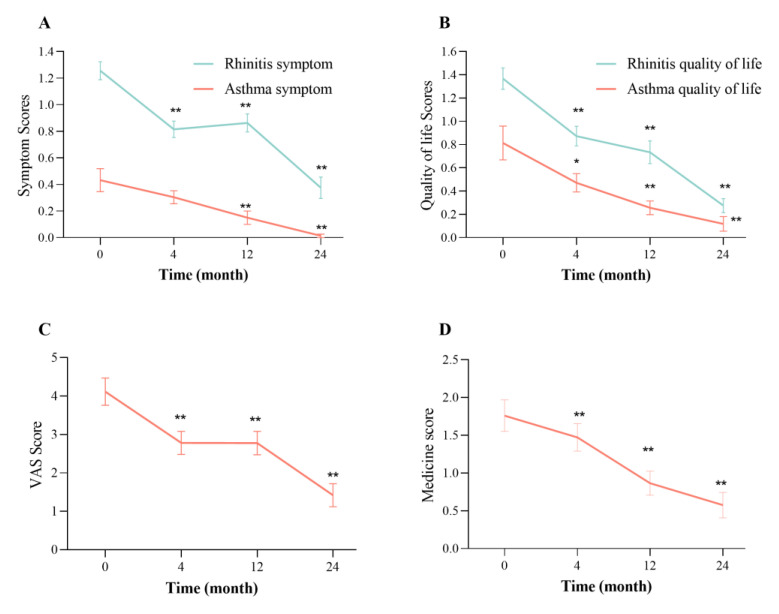
Clinical symptoms improved after AIT. The symptom scores (**A**), quality of life scores (**B**), VAS score (**C**), and medicine score (**D**) were significantly improved at 4, 12, and 24 months of AIT compared with the baseline values (* *p* < 0.05, ** *p* < 0.01), displaying a continuous downward trend.

**Figure 2 jcm-12-01665-f002:**
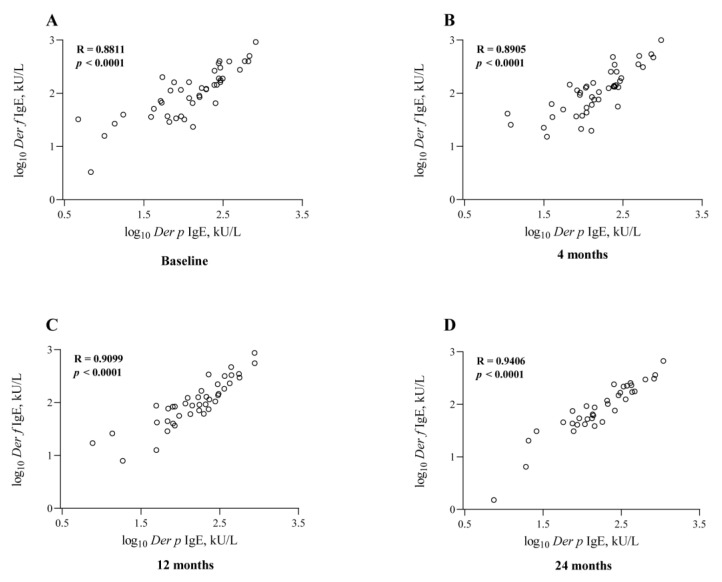
Serum *Der p*-specific IgE levels were closely associated with serum *Der f*-specific IgE at the baseline (**A**), 4 (**B**), 12 (**C**), and 24 (**D**) months (*p* < 0.001) after the standardizing transform, suggesting a cross-reaction between them.

**Figure 3 jcm-12-01665-f003:**
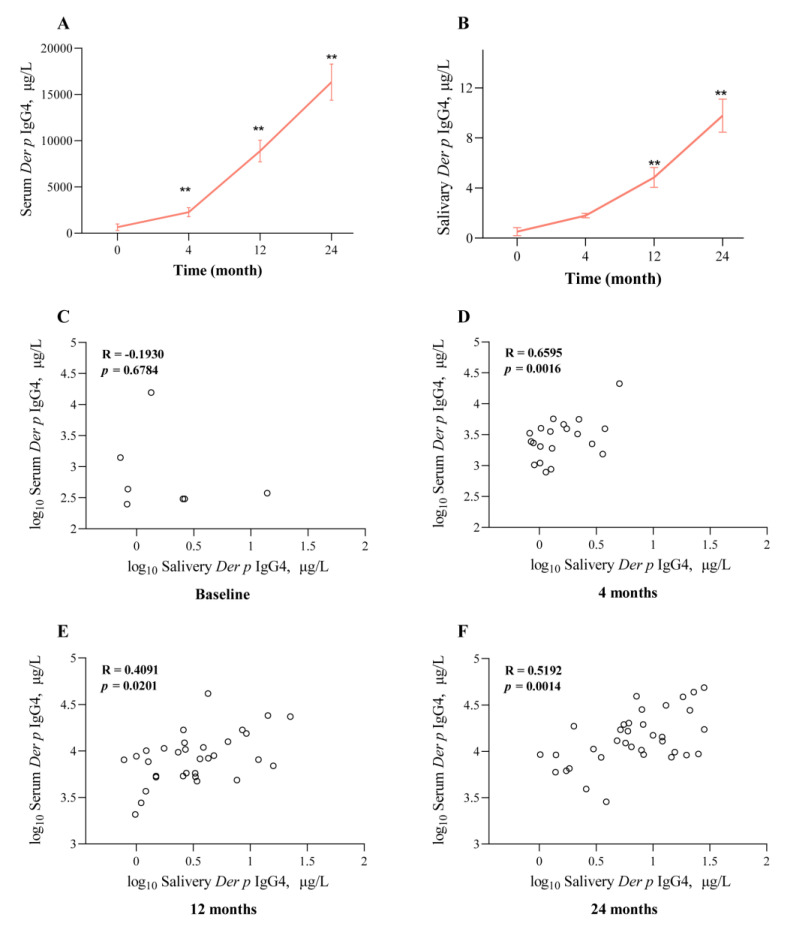
Serum (**A**) and salivary (**B**) *Der p*-specific IgG4 significantly increased with the time of AIT treatment. There was no obvious correlation between *Der p*-specific IgG4 in serum and saliva before AIT treatment (**C**). Notably, significant correlations between them were detected at 4 (**D**), 12 (**E**), and 24 (**F**) months of AIT. ** *p* < 0.01.

**Figure 4 jcm-12-01665-f004:**
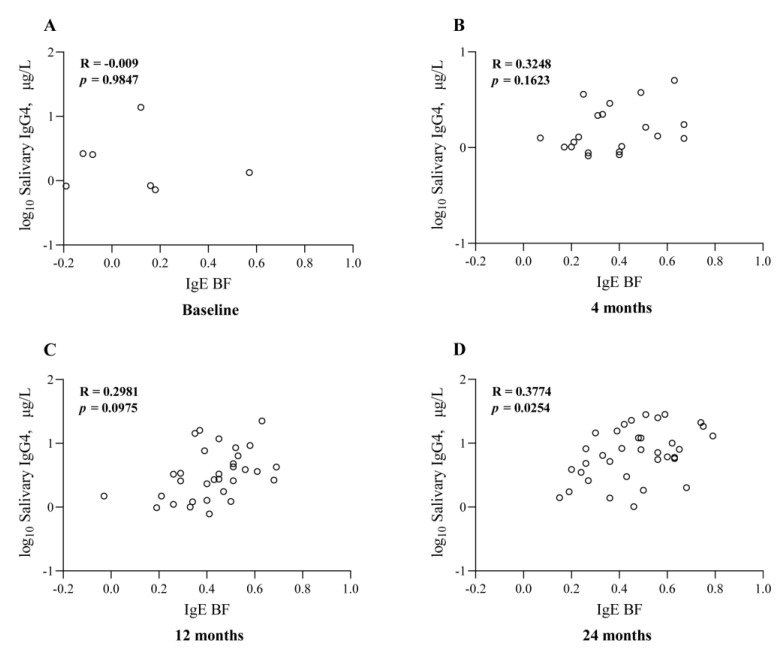
Significant correlations between serum *Der p*-specific IgE-BF and *Der p*-specific IgG4 at the baseline (**A**), 4 (**B**), 12 (**C**), and 24 (**D**) months after AIT.

**Figure 5 jcm-12-01665-f005:**
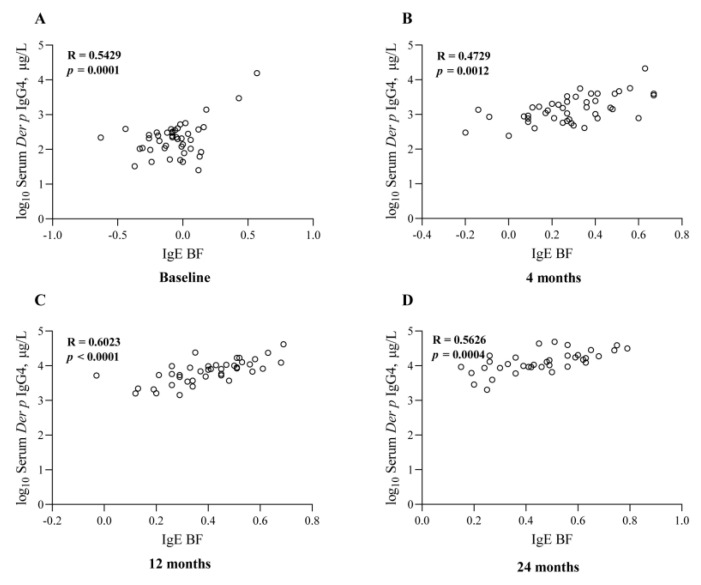
Salivary IgG4 was detected only in a small number of patients before treatment, and no significant correlation was observed between salivary *Der p*-specific IgG4 and serum-specific IgE-BF (**A**). However, after AIT began, their correlations gradually increased, reaching the maximum at 24 months of AIT (**B**–**D**).

**Table 1 jcm-12-01665-t001:** Clinical characteristics of included children with asthma and/or allergic rhinitis.

Characteristics before Treatment	AIT Group (*n* = 44)	Control Group (*n* = 11)	*p*-Value
Age (years), Mean (SD)	7.11 (2.04)	6.83 (2.23)	0.78
Gender (male/female, n/n)	34/10	6/5	0.15
Diagnosis (AR/AR+Asthma, n/n)	15/28	4/7	0.99
*Der p* IgE, Median (IQR), kU/L	145.18 (65.49–280.63)	46.81 (5.32–120.49)	0.20
*Der f* IgE, Median (IQR), kU/L	117.38 (39.04–193.00)	29.67 (2.15–99.13)	0.20
*Der p* IgE-BF, Median (IQR)	−0.05 (−0.18–0.02)	0.13 (−0.02–0.23)	0.09
*Der f* IgE-BF, Median (IQR)	−0.09 (−0.21–−0.02)	−0.02 (−0.06–0.03)	0.03 *
Salivary IgG4, Mean (SD), ug/L	0.52 (2.14)	0.45 (1.50)	0.38
Serum IgG4, Median (IQR), ug/L	213.14 (104.80–328.87)	144.40 (74.86–214.62)	0.55
Rhinitis symptom score, Median (IQR)	1.25 (1.00–1.50)	1.25 (0.88–2.00)	0.71
Asthma symptom score, Median (IQR)	0.25 (0–0.50)	0.75 (0.50–1.25)	0.02 *
VAS symptom score, Median (IQR)	4.00 (3.00–5.25)	4.00 (3.00–5.00)	0.93
Medicine score (A week), Median (IQR)	2.00 (0–3.00)	0.43 (0–2.86)	0.55
Rhinitis quality of life score, Mean (SD)	1.37 (0.61)	1.14(1.13)	0.55
Asthma quality of life score, Mean (SD)	0.81 (0.96)	1.16 (0.88)	0.16

AIT, allergen immunotherapy; AR, allergic rhinitis; BF, blocking factor; IQR, interquartile range; SD, standard deviation; VAS, visual analog scale; * *p* < 0.05.

**Table 2 jcm-12-01665-t002:** Laboratory and clinical characteristics of children with asthma and/or AR after AIT.

The Change from Baseline in AIT Group	Baseline	4 Months	12 Months	24 Months
*Der p* IgE, Median (IQR), kU/L	145.18 (65.49–−280.63)	134.85 (90.93–257.85)	174.3 (80.60–300.30)	162.77 (90.47–362.99)
*Der f* IgE, Median (IQR), kU/L	117.38 (39.04–−193.00)	118.75 (52.55–175.13)	92.60 (60.50–182.90)	81.80 (44.93–186.34)
*Der p* IgE-BF, Median (IQR)	−0.05 (−0.18–−0.02)	0.28 (0.16–0.40) **	0.40 (0.29–0.51) **	0.47 (0.32–0.59) **
*Der f* IgE-BF, Median (IQR)	−0.09 (−0.21–−0.02)	0.24 (0.05–0.41) **	0.35 (0.18–0.51) **	0.40 (0.21–0.51) **
Salivary IgG4, Mean (SD), ug/L	0.52 (2.14)	1.79 (1.17)	4.84 (5.04) **	9.78 (8.00) **
Serum IgG4, Median (IQR), ug/L	213.14 (104.80–328.87)	1393.58 (769.88–2642.78) **	7692.86 (4733.20–10,429.23) **	12598.40 (9169.74–19,524.03) **
Rhinitis symptom score, Median (IQR)	1.25 (1.00–1.50)	0.75 (0.50–1.00) **	1.00 (0.75–1.15) **	0 (0–0.75) **
Asthma symptom score, Median (IQR)	0.25 (0–0.50)	0.25 (0–0.50)	0 (0–0.25) **	0 (0) **
VAS symptom score, Median (IQR)	4.00 (3.00–5.25)	2.00 (2.00–4.00) **	3.00 (1.00–4.00) **	1.00 (0–2.00) **
Medicine score (A week), Median (IQR)	2.00 (0–3.00)	1.36 (0.43–2.00)	0.29 (0–2.00) **	0 (0–1.00) **
Rhinitis quality of life score, Mean (SD)	1.37 (0.61)	0.87 (0.56) **	0.73 (0.63) **	0.27 (0.36) **
Asthma quality of life score, Mean (SD)	0.81 (0.96)	0.47 (0.52) *	0.26 (0.38) **	0.12 (0.38) **

AIT, allergen immunotherapy; AR, allergic rhinitis; SD, standard deviation; IQR, interquartile range; VAS, visual analog scale; * *p* < 0.05 and ** *p* < 0.01.

**Table 3 jcm-12-01665-t003:** Clinical characteristics of control children with asthma/AR.

The Change from Baseline in Control Group	Baseline	12 Months	*p*-Value
*Der p* IgE, Median (IQR), kU/L	46.81 (5.32–120.49)	68.20 (5.15–148.25)	0.85
*Der f* IgE, Median (IQR), kU/L	29.67 (2.15–99.13)	19.55 (2.68–44.86)	0.60
*Der p* IgE-BF, Median (IQR)	0.13 (−0.02–0.23)	0.13 (0.06–0.26)	0.64
*Der f* IgE-BF, Median (IQR)	−0.02 (−0.06–0.03)	0 (−0.11–0.17)	0.95
Salivary IgG4, Mean (SD), ug/L	0.45 (1.50)	0.60 (1.21)	0.39
Serum IgG4, Median (IQR), ug/L	144.40 (74.86–214.62)	144.40 (74.86–214.62)	0.55
Rhinitis symptom score, Median (IQR)	1.25 (0.88–2.00)	1.87 (1.50–2.00)	0.69
Asthma symptom score, Median (IQR)	0.75 (0.50–1.25)	0.27 (0.25–0.31)	0.06
VAS symptom score, Median (IQR)	4.00 (3.00–5.00)	4.00 (2.50–5.25)	0.95
Medicine score (A week), Median (IQR)	0.43 (0–2.86)	0.50 (0–1.50)	0.74
Rhinitis quality of life score, Mean (SD)	1.14 (1.13)	1.12 (0.63)	0.97
Asthma quality of life score, Mean (SD)	1.16 (0.88)	0.69 (0.66)	0.30

AIT, allergen immunotherapy; AR, allergic rhinitis; SD, standard deviation; IQR, interquartile range; VAS, visual analog scale.

## Data Availability

Not applicable.

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
