# Peer review of "Salivary IgG4 Levels Contribute to Assessing the Efficacy of Dermatophagoides pteronyssinus Subcutaneous Immunotherapy in Children with Asthma or Allergic Rhinitis"

_jcm, 2023, doi:10.3390/jcm12041665_

Round 1

Reviewer 1 Report

Thank you very much for giving me an opportunity to review the manuscript.

Authors conducted two-years long prospective AIT study and presented salivary IgG4 trend.

Major points

In discussion section, authors mentioned the positive correlation between serum and salivary specific IgG4 levels to allergens further indicated that IgG4 could be passively transferred rather than locally secreted with refference8. Therefore, authors’ question for conducting this study is already reported. What is difference between this manuscript and the reference?

Author showed close relationship between serum and saliva IgG4. However, authors did not directly see the effectiveness of AIT and saliva IgG4. If IgG4 is a direct or surrogate marker of clinical response of AIT, please compare them directly.

Authors mentioned repeatedly that AIT in children is very effective with references. If it is true, is there any needs for biomarker for AIT in children?

If saliva IgG4 is biomarker for clinical responses, what is cut-off value for good response?

Please show the difference in saliva IgG4 value in both responder and non-responders.

Minor points

“Asthma and/or AR” and “AR and/or asthma” are used in text. Is there any difference? And also “IgE -BF” and “IgE BF” are same case.

In Tables some IgE value is expressed in minus values. IgE value should be over zero.

Author Response

Dear editor,

We carefully studied reviewer’s comments and have made the necessary changes according to the suggestions. Our responses to reviewer’s comments are listed below:

Reviewer 1: Major points

1.1 In discussion section, authors mentioned the positive correlation between serum and salivary specific IgG4 levels to allergens further indicated that IgG4 could be passively transferred rather than locally secreted with reference 8. Therefore, authors’ question for conducting this study is already reported. What is difference between this manuscript and the reference?

Response: This is a good question. In discussion section, we mentioned the positive correlation between serum and salivary specific IgG4 levels to allergens further indicated that IgG4 could be passively transferred rather than locally secreted with reference 8. For this cited article, Miranda et al. only enrolled those children with allergic rhinitis (AR) with or without mild-to-moderate asthma, who did not received allergen immunotherapy (AIT). Different from their study, our study focused on the immune response of children with asthma/AR after receiving AIT. Furthermore, we followed up for 2 years to further study antibody reaction after AIT.

1.2 Author showed close relationship between serum and saliva IgG4. However, authors did not directly see the effectiveness of AIT and saliva IgG4. If IgG4 is a direct or surrogate marker of clinical response of AIT, please compare them directly. Response: In the present study, included children with asthma and/or AR were mild-to-moderate. The correlations between saliva IgG4 and clinical scores were limited (Symptom score R = 0.054, P = 0.142; Quality of life score R < 0.0001, P = 0.9814; VAS score R = 0.01353, P = 0.4689; Medical score R = 0.01153, P = 0.5039). This result may be related to the following reasons. Their initial scores are not very high, and the improvements are not too obvious after AIT. On the other hand, our sample size relatively small. Therefore, it is necessary to conduct a large sample, multi-center clinical study to further explore their correlation. Notably, the overall line chart can indicate a significant improvement trend.

1.3 Authors mentioned repeatedly that AIT in children is very effective with references. If it is true, is there any needs for biomarker for AIT in children?

Response: So far, the clinical efficacy evaluation of AIT includes assessing their symptoms, recording the medication use, and filling out the quality-of-life questionnaire. However, these evaluation indicators are subjective and may lead to bias. Therefore, it is necessary to use some objective indicators to evaluate the efficacy of AIT. In our study, salivary specific IgG4 levels are used as a biomarker for evaluating the efficacy of subcutaneous AIT. Furthermore, salivary specific IgG4 levels are closely associated with serum IgG4 levels and clinical scores. This non-invasive objective salivary IgG4 detection method may greatly increase children's medical experience, and improve patient’s compliance, possibly being suitable for children.

1.4 If saliva IgG4 is biomarker for clinical responses, what is cut-off value for good response?

Response: At the 24th month, the cutoff value of saliva IgG4 for protective antibodies was 2.006, and its sensitivity and specificity were 0.83 and 0.77 respectively. We have added this content in Results.

1.5 Please show the difference in saliva IgG4 value in both responder and non-responders.

Response: The included patients were divided into the responder (high protective antibodies) and non-responder (low protective antibodies) groups based on the cutoff value at the 24th month. There was significant difference between the responder and non-responder groups (11.17 ±7.88 vs 1.23 ± 0.01, P < 0.001).

Minor points:

1.6 “Asthma and/or AR” and “AR and/or asthma” are used in text. Is there any difference? And also “IgE-BF” and “IgE BF” are same case.

Response: we made the corresponding revision. “Asthma and/or AR” and “AR and/or asthma” was uniformly changed to asthma and/or AR. “IgE-BF” and “IgE BF” were uniformly changed to IgE-BF.

1.7 In Tables some IgE value is expressed in minus values. IgE value should be over zero.

Response: The IgE value in our Tables are expressed in positive values. However, the values for IgE-BF were standardized, ranging from -1 to +1. 

We have addressed all the suggestions of reviewer. The modifications in the manuscript were shown in red. Undoubtedly, the incorporation of these comments has improved the quality and clarity of this manuscript. We thank again you for your consideration of this manuscript.

Sincerely yours,

Prof. Xuefeng Xu,

Reviewer 2 Report

It is an interesting article. However, some points would need consideration:

1.       The first sentence in the abstract “At present, there was no effective non-invasive detection to evaluate the efficacy of pediatric house dust mite (HDM)-specific allergen immunotherapy (AIT)” needs to be rephrased. Currently, the only accepted criteria to evaluate the efficacy of allergen immunotherapy is the symptom and medication score, whic is a non-invasive method.

2.       Lines 93 to 95: the 30 min after each administration is not to avoid possible adverse reactions, is for detecting and treating any immediate adverse reaction.

3.       Section 2.3 (Clinical evaluations).

·         The authors explain that the patients were requested to assess the symptoms and record medication over the past weeks at each visit. ¿How many weeks?

·      What RQLQ and AQLQ were used? Please, describe.

4.     Section 2.5. Please, describe the method for the measurement of IgG4 and how the results are expressed. 

Author Response

Dear editor,

We carefully studied reviewer’s comments and have made the necessary changes according to the suggestions. Our responses to reviewer’s comments are listed below:

Reviewer 2:

2.1 The first sentence in the abstract “At present, there was no effective non-invasive detection to evaluate the efficacy of pediatric house dust mite (HDM)-specific allergen immunotherapy (AIT)” needs to be rephrased. Currently, the only accepted criteria to evaluate the efficacy of allergen immunotherapy is the symptom and medication score, which is a non-invasive method.

Response: This is a good suggestion. We made corresponding revision “At present, there was no effective, non-invasive, and objective indicators to evaluate the efficacy of pediatric house dust mite (HDM)-specific allergen immunotherapy (AIT)”.

2.2 Lines 93 to 95: the 30 min after each administration is not to avoid possible adverse reactions, is for detecting and treating any immediate adverse reaction.

Response: This is a good suggestion. We made corresponding revision.

2.3 Section 2.3 (Clinical evaluations). The authors explain that the patients were requested to assess the symptoms and record medication over the past weeks at each visit. ¿How many weeks?

Response: The patients were requested to assess the symptoms and record medication over the past week at each visit. In general, patients need to record their symptoms within one week.

2.4 What RQLQ and AQLQ were used? Please, describe.

Response: RQLQ refers to rhinoconjunctivitis quality of life questionnaire, and AQLQ refers to asthma quality of life questionnaire. RQLQ is composed of 7 domains, including activity limitation, sleep, non-nose/eye symptoms, practical problems, nose symptoms, eye symptoms, and emotional function with 28 questions; and AQLQ included 31 questions, involving in activity limitation, symptoms, emotional function, and exposure to environmental stimuli. We have made revision in the section of Method and added the references.

2.5 Section 2.5. Please, describe the method for the measurement of IgG4 and how the results are expressed.

Response: We have added the following description “In brief, the microplate was coated with 10 μg/mL Der p extract (ALK-Abello, Denmark) protein. Then blocked with 2% casein, and subsequently incubated with diluted standards, controls, and serum samples. Mouse monoclonal antibodies to human IgG4 (ALK-Abello, Denmark) was used at 1/10,000 for detection, and revealed by HRP labeled goat anti-mouse IgG (1/20,000; KPL, USA), then followed by Tetramethylbenzidine visualizing. The assays were further calibrated against ImmunoCAP-specific IgG4 (Uppsala, Sweden) by measuring the specific IgG4 concentration in mgA/L of Der p IgG4 in the ELISA standard stock solutions by ImmunoCAP. The lower limit of quantification (LoQ) of serum Der p IgG4 was 3.33 μg/L, and the LoQ of salivary Der p IgG4 was 0.67. Serum samples needed to be diluted at least 5-fold due to the different matrix effects, whereas there was no need to dilute saliva samples.”

We have addressed all the suggestions of reviewer. The modifications in the manuscript were shown in red. Undoubtedly, the incorporation of these comments has improved the quality and clarity of this manuscript. We thank again you for your consideration of this manuscript.

Sincerely yours,

Prof. Xuefeng Xu

Round 2

Reviewer 1 Report

Authors responded my concern point by point. However, Response to major point1.1 is not shown in main text. Please include your response in discussion section. 

Author Response

Dear editor,

We carefully studied reviewer’s comment and have made the necessary change according to the suggestion. Our response to reviewer’s comment is listed below:

Reviewer 1: Authors responded my concern point by point. However, Response to major point1.1 is not shown in main text. Please include your response in discussion section.

Response: In discussion section, we added the corresponding content “Different from this study by Miranda et al. that they only enrolled those AR and/or asthma children who did not receive AIT [8], our study focused on the immune responses of children with asthma/AR after receiving AIT. Furthermore, we followed up for 2 years to further study antibody reactions after AIT.”.

We have made the corresponding revision, and the added content was shown in red in section of Discussion. Undoubtedly, the incorporation of the comment has improved the quality and clarity of this manuscript. We appreciate the reviewers’ constructive comments, and we thank again you for your consideration of this manuscript.

Sincerely yours,

Prof. Xuefeng Xu